# MAPK-mediated transcription factor GATAd contributes to Cry1Ac resistance in diamondback moth by reducing *PxmALP* expression

Le Guo[1,2,3☯], Zhouqiang Cheng[1,3☯], Jianying Qin[3☯], Dan Sun[3☯], Shaoli Wang[3], Qingjun Wu[3], Neil Crickmore[4], Xuguo Zhou[5], Alejandra Bravo[6], Mario Soberón[6], Zhaojiang Guo[3]*, Youjun Zhang[3]*

**1** College of Plant Protection, Hunan Agricultural University, Changsha, China, **2** Guangdong Laboratory for Lingnan Modern Agriculture, Guangdong, China, **3** Department of Plant Protection, Institute of Vegetables and Flowers, Chinese Academy of Agricultural Sciences, Beijing, China, **4** School of Life Sciences, University of Sussex, Brighton, United Kingdom, **5** Department of Entomology, University of Kentucky, Lexington, Kentucky, United States of America, **6** Departamento de Microbiología Molecular, Instituto de Biotecnología, Universidad Nacional Autónoma de México, Cuernavaca, Morelos, México

☯ These authors contributed equally to this work.
* guozhaojiang@caas.cn (ZG); zhangyoujun@caas.cn (YZ)

**Data Availability Statement:** All relevant data are within the manuscript and its Supporting Information files. The final cloned full-length cDNA sequence of P. xylostella GATAd gene was

## Abstract

The benefits of biopesticides and transgenic crops based on the insecticidal Cry-toxins from *Bacillus thuringiensis* (Bt) are considerably threatened by insect resistance evolution, thus, deciphering the molecular mechanisms underlying insect resistance to Bt products is of great significance to their sustainable utilization. Previously, we have demonstrated that the down-regulation of *PxmALP* in a strain of *Plutella xylostella* (L.) highly resistant to the Bt Cry1Ac toxin was due to a hormone-activated MAPK signaling pathway and contributed to the resistance phenotype. However, the underlying transcriptional regulatory mechanism remains enigmatic. Here, we report that the PxGATAd transcription factor (TF) is responsible for the differential expression of *PxmALP* observed between the Cry1Ac susceptible and resistant strains. We identified that PxGATAd directly activates *PxmALP* expression via interacting with a non-canonical but specific GATA-like *cis*-response element (CRE) located in the *PxmALP* promoter region. A six-nucleotide insertion mutation in this *cis*-acting element of the *PxmALP* promoter from the resistant strain resulted in repression of transcriptional activity, affecting the regulatory performance of PxGATAd. Furthermore, silencing of *PxGATAd* in susceptible larvae reduced the expression of *PxmALP* and susceptibility to Cry1Ac toxin. Suppressing *PxMAP4K4* expression in the resistant larvae transiently recovered both the expression of *PxGATAd* and *PxmALP*, indicating that the PxGATAd is a positive responsive factor involved in the activation of *PxmALP* promoter and negatively regulated by the MAPK signaling pathway. Overall, this study deciphers an intricate regulatory mechanism of *PxmALP* gene expression and highlights the concurrent involvement of both *trans*-regulatory factors and *cis*-acting elements in Cry1Ac resistance development in lepidopteran insects.

deposited in the GenBank database (accession no. MZ712004). The raw data of the figures and statistical analyses in this study are provided in S1 Data.

**Funding:** This work was supported by the Laboratory of Lingnan Modern Agriculture Project (NT2021003) to YZ, the National Natural Science Foundation of China (32022074; 32172458) to ZG, the Beijing Key Laboratory for Pest Control and Sustainable Cultivation of Vegetables and the Science and Technology Innovation Program of the Chinese Academy of Agricultural Sciences (CAAS-ASTIP-IVFCAAS) to YZ. The funders had no role in study design, data collection and analysis, decision to publish, or preparation of the manuscript.

**Competing interests:** The authors have declared that no competing interests exist.

## Author summary

Gene expression and regulation are associated with adaptive evolution in living organisms. The rapid evolution of insect resistance to Bt insecticidal Cry toxins is frequently associated with reduced expression of diverse midgut genes that code for Cry-toxin receptors. Nonetheless, our current knowledge about the regulation of gene expression of these pivotal receptor genes in insects is limited. Membrane-bound alkaline phosphatase (mALP) is a known receptor for Cry1Ac toxin in diverse insects and here, we report the transcriptional regulatory mechanism of the *PxmALP* gene related to Cry1Ac resistance in *P. xylostella*. We identified a MAPK signaling pathway that negatively regulates the PxGATAd transcriptional factor which is involved in the differential expression of *PxmALP* via interacting with the *PxmALP* promoter. Furthermore, a *cis*-acting element mutation repressing the regulatory activity of PxGATAd for *PxmALP* expression in the Cry1Ac resistant strain was identified. Our study provides an insight into the precise transcriptional regulatory mechanism that regulates *PxmALP* expression and is involved in the evolution of Bt Cry1Ac resistance in *P. xylostella*, which provides a paradigm for decoding the regulation landscape of midgut Cry-toxin receptor genes in insects.

## Introduction

The potential of insecticide application is fading away worldwide, as their efficacy against insect pests is diminished by the evolution of resistance [1]. *Bacillus thuringiensis* (Bt) is an entomopathogenic bacterium that represents the most successful biopesticide [2]. To date, formulations with Bt toxins and transgenic crops expressing Bt insecticidal toxin genes have been widely adopted for pest management worldwide, providing unprecedented economic, ecological, and social benefits [3–7]. However, the ability of insects to adapt to Bt toxins supports the conclusion that the rapid evolution of resistance by insects is a major threat for the future application and success of Bt-based products [8, 9]. Hence, it is important to decipher the molecular mechanisms underlying insect resistance to Bt toxins for their sustainable utilization. The mode of action of Bt toxins implies interactions between toxins and larval midgut proteins that function as receptors [10, 11]. The midgut receptors of Bt toxins include proteins such as cadherin-like (CAD), aminopeptidase-N (APN), alkaline phosphatase (ALP) and ATP-binding cassette (ABC) transporters [12]. Besides gene mutations in these proteins, accumulating evidence suggest that their reduced expression is also strongly associated with high levels of insect resistance to Cry proteins [13]. Nevertheless, the regulatory mechanisms involved in the expression of these midgut receptors are unclear.

Existing evidence indicates that differences in gene expression can arise from *cis*-regulatory mutations affecting gene transcription, or from mutations affecting *trans*-regulatory elements involved in gene regulation [14]. Gene regulatory networks evolve by the coevolution of transcription factors (TFs) and interaction with their target *cis*-regulatory sequences [15]. Accordingly, both *cis*- and *trans*-regulatory changes lead to divergent gene expression landscapes [16]. Transcriptional regulation of genes is a vital contributor for the evolutionary adaptation to environmental changes, where *cis* and *trans* elements involved in gene expression play an important role in dynamic environmental adaptation [17]. A considerable body of evidence has demonstrated that both *cis*-regulatory elements [18] and TFs are powerful players involved in the molecular and genetic mechanisms participating in insect metamorphosis [19, 20] and in behavioural plasticity [21]. The overexpression of detoxifying enzyme genes associated with

metabolic resistance can be achieved through changes in *cis*-acting promoter regions and/or *trans*-acting regulatory loci [22, 23]. With advances in bioinformatics, the causal genetic variants that promote the development of quantitative traits, such as pesticide resistance, are now being illustrated in arthropod pests [24].

Spatio-temporal specific patterns of transcription are determined by the global interaction of diverse TFs with *cis*-acting elements, such as enhancers or negative regulatory *cis* elements [25–27]. A substantial fraction of genetic variation in the promoter sequences has consequences for adaptation, where the promoter sequences are more variable than the coding regions [28, 29]. For instance, sequence variations in core promoter regions result in differential gene expression levels in *Drosophila melanogaster*, providing diversity and impetus for natural selection [30]. *Cis*-acting mutations detected in the promoter region, contribute to altered expression of genes that could lead to resistance phenotypes [31–33] and can act by affecting transcription by altering the efficiency of TF action [34].

*Plutella xylostella* is one of the most voracious and devastating lepidopteran pests in the agricultural production of brassicaceous vegetables around the world and was the first insect to develop resistance to Bt toxin-based biopesticides, providing a good model for exploring the resistance mechanisms to Bt [35]. In general, soluble ALP and membrane-bound ALP (mALP) are involved in dephosphorylation reactions at basic pH values in the insect gut. In the case of mALPs, it has been shown that this protein serves as binding site for Bt Cry proteins increasing the toxin concentration at the epithelial surface of larval midgut cells in a variety of insects [12]. We previously showed that the hormone-activated MAPK signal pathway negatively regulates the expression of *PxmALP*, and other receptors, in the Cry1Ac-resistant NIL-R strain of *P. xylostella* conferring resistance to Bt Cry1Ac toxin [36–39]. However, it still remains to be defined how MAPK affects the transcriptional regulation of these receptors.

With the aim of investigating the molecular basis of the differential expression of *PxmALP* in *P. xylostella*, we analyzed the contribution of *trans*- and *cis*-acting elements potentially involved in *PxmALP* gene expression. Our results unmasked a regulatory network where a PxGATAd *trans*-regulatory element interacted with a *cis*-acting element identified in the promoter of *PxmALP*, modulating the expression of *PxmALP*. A mutation was identified in this element in the resistant strain which reduced PxGATAd binding and consequently *PxmALP* expression. This study provides a model for better comprehending the impacts of transcriptional regulation on shaping the expression atlas of midgut genes within the context of combating Bt Cry toxins in insect pests.

## Results

### Analysis of the 5′-flanking region of *PxmALP* in susceptible and resistant strains

Previous work has indicated that in *P. xylostella* the down-regulation of *PxmALP* and other receptors is linked to Cry1Ac resistance in the NIL-R strain [36]. To analyze if the transcriptional down-regulation of *PxmALP* in the NIL-R resistant strain, compared to the DBM1Ac-S susceptible strain, was due to differences in its promoter region, genomic DNA templates prepared from individuals from both strains were used to amplify the 5′-flanking sequences of *PxmALP*. Sequence analysis revealed that the promoter sequences showed multiple differences between susceptible and resistant individuals (S1 Fig). The DNA sequences showed a conserved transcription initiation site (Inr) motif 5′-TCAG-3′ located 328 bp upstream of the translation start site (S1 Fig). A total of 115 SNP and sequence indel (insertion/deletion) differences were detected when the two major alleles were compared. The full-length sequences were 2004 and 2296 bp for the resistant and susceptible strains, respectively (Fig 1A). This size

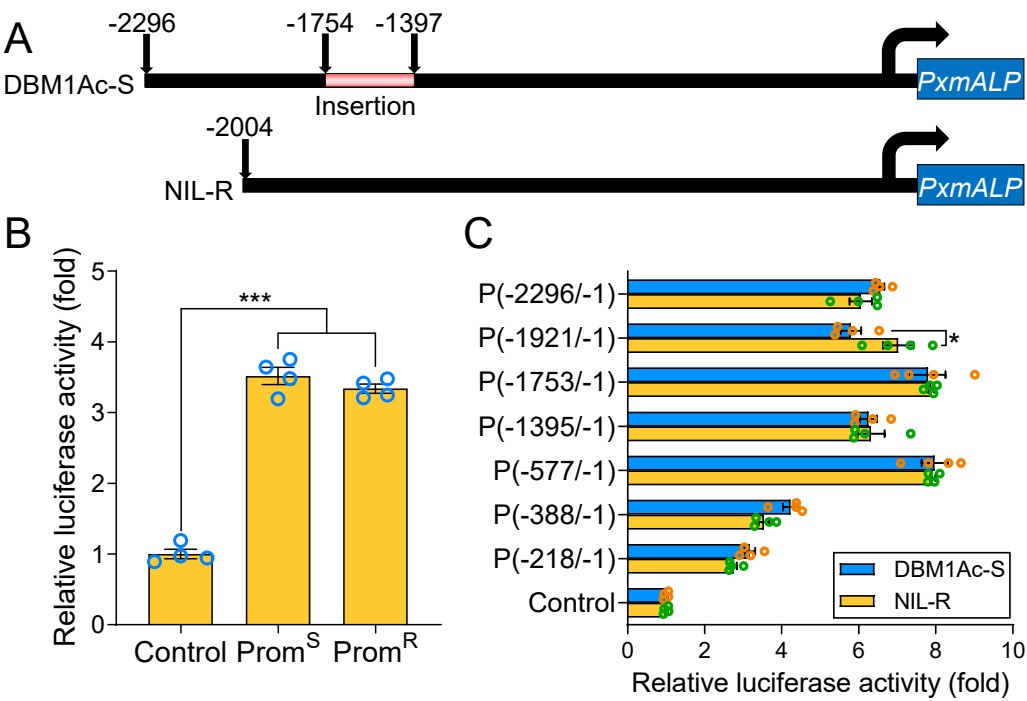

**Fig 1. Analysis of *PxmALP* promoter activity, in DBM1Ac-S susceptible and NIL-R resistant *P. xylostella* strains.** (A) Diagram of the promoter of the *PxmALP* gene. An insertion is represented by the pink rectangle in the promoter of the *PxmALP* allele from the DBM1Ac-S strain. The right-angled arrow denotes the transcriptional start site (TSS). (B) Detection of the transcriptional activity by dual-luciferase activity reporter assay system (pGL4.10 plasmid was used as control and normalized the values as 1-fold) of *PxmALP* promoters cloned from the susceptible (Prom^S) and resistant (Prom^R) *P. xylostella* strains. (C) Promoter activity analysis performed on progressive 5′-end deletions of the flanking region of *PxmALP*. The name of each fragment construction is composed of a "P" and two numerals separated by a dash within parentheses, to specify the 5′ and 3′ positions of the corresponding promoter fragment from the susceptible strain. (B and C) The data and error bars represent the mean and standard errors of the mean (SEM) by four biological replicates. The value of relative luciferase activity (Firefly luciferase activity/Renilla luciferase activity) was normalized to the empty pGL4.10 vector. The significance of the observed differences was determined by one-way ANOVA with Holm-Sidak's test ($^*p < 0.05$, $^{***}p < 0.001$).

difference was largely due to a discrete fragment (264 bp) located between -1397 to -1754 bp in the susceptible sequence but absent in the resistant sequence (Fig 1A). The sequences from both strains were used to construct luminescent reporter plasmids that were transfected into S2 insect cell line and were found to show similar promoter activity when expressed in S2 cells (Fig 1B).

To identify regions in the in the upstream sequence that are required for transcription we decided to clone different *PxmALP*-flanking region fragments from the susceptible and resistant strains and compared their promoter activity in luminescent reporter plasmids. Deletions down to -577 had little effect on transcription while larger deletions reduced expression, albeit not as far as the control construct lacking any promoter sequence (Fig 1C). No differences, however, were seen between the susceptible and resistant sequences.

## PxGATAd directly up-regulates *PxmALP* transcription

Since we could not identify any differences in the basal promoter activity between sequences from susceptible and resistant strains, we considered whether TFs may play a role in modulating the transcriptional activity of *PxmALP* in the NIL-R resistant strain. To identify possible TF candidates responsible for *PxmALP* gene expression, we predicted putative *cis*-response

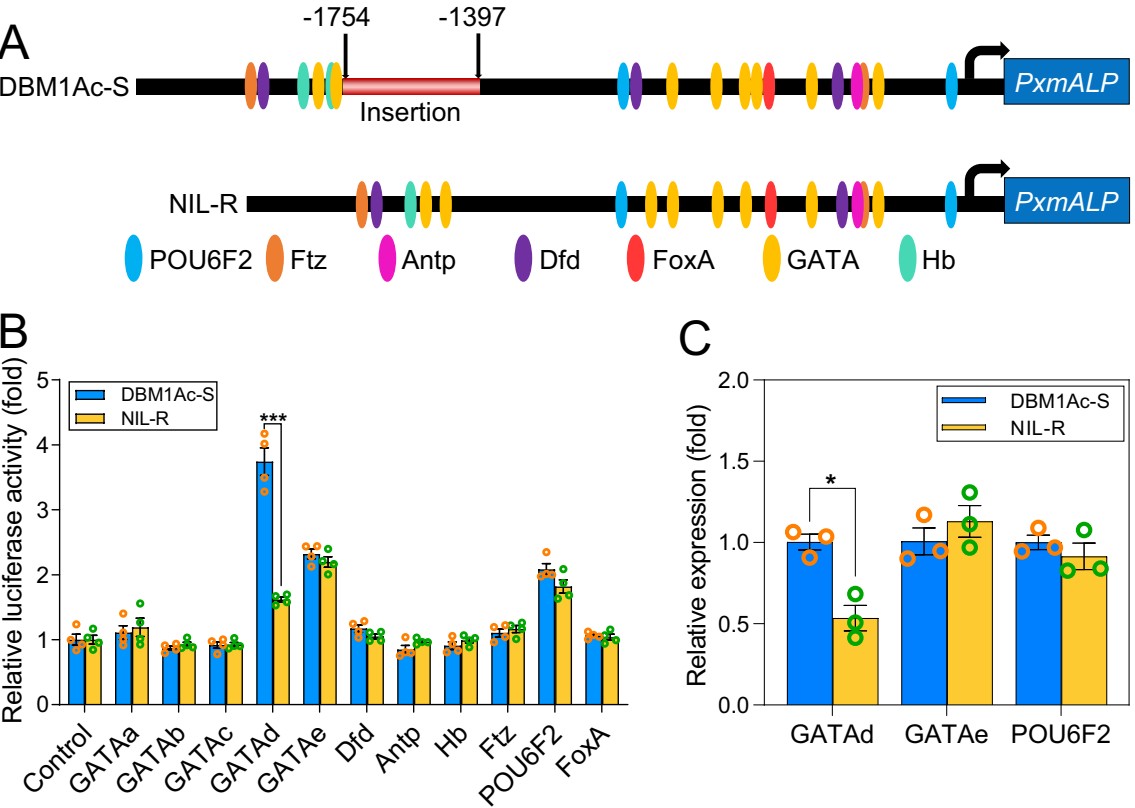

**Fig 2. Role of different TFs on the transcriptional activation of the *PxmALP* promoter.** (A) Schematic illustration of the putative TF CREs located in the promoter region of *PxmALP*. Different CREs are indicated by colored ellipses. (B) Analysis of various TFs on the activity of the *PxmALP* promoter. The values shown are the means and the corresponding SEM values. The empty vector pAc5.1/V5-His B was used as control and standardized the value as 1-fold. Four independent transfections were conducted for each pair of plasmids. Holm-Sidak's test was used for statistical analysis (***$p < 0.001$). (C) The expression of *PxGATAd*, *PxGATAe* and *PxPOU6F2* was analyzed in the midgut tissue from larvae of DBM1Ac-S and NIL-R strains. The values are presented as mean of relative expression values ± the SEM (*$p < 0.05$, Holm-Sidak's test, n = 3).

elements (CREs) in the promoter of *PxmALP* by using two different *in silico* methods, ALL-GEN program [40, 41] and JASPAR database [42]. Several CREs were predicted in the promoter region of *PxmALP* that may be candidates to respond to TFs, like GATA, Dfd, Antp, Hb, Ftz, POU6F2, and FoxA (Fig 2A). To identify if any of the predicted TFs regulate *PxmALP* expression, five GATA family members GATAa, GATAb, GATAc, GATAd and GATAe, and six other TFs were cloned and cotransfected into S2 cells with the promoter constructs from both NIL-R and DMB1Ac-S strains fused to a luminescent reporter. We first analyzed the activity of the promoter region cloned from the DBM1Ac-S susceptible strain in combination with the different TFs. The luciferase assays showed a significant increase in the transcriptional activity driven by PxGATAd, and a slight increase was also observed in conjunction with PxGATAe and PxPOU6F2 (Fig 2B). Interestingly, when we analyzed the activity of the promoter region cloned from the NIL-R strain the data showed that transcriptional activity induced by PxGATAd was significantly different, displaying lower activity when compared to DBM1Ac-S, while no differences were observed with PxGATAe and PxPOU6F2 (Fig 2B). We then conducted qPCR to establish whether the transcript levels of these three TFs varied between the resistant and susceptible strains. The results indicated that the expression level of PxGATAd in the susceptible DBM1Ac-S strain was significantly higher than in the resistant

NIL-R strain. However, PxGATAe and PxPOU6F2 displayed similar expression levels (Fig 2C). These results supported the idea that PxGATAd, in contrast to PxGATAe or PxPOU6F2, is a key factor in controlling the differential expression of *PxmALP* between *P. xylostella* NIL-R and DBM1Ac-S strains.

## PxGATAd enhanced the activity of the *PxmALP* promoter through non-canonical GATA-like CREs

The GATAd protein contains a conserved GATA-type zinc finger (ZnF) domain (S2 Fig). A phylogenetic analysis confirmed that the GATAd proteins are evolutionarily conserved and clearly clustered with the corresponding proteins from the same insect order (S3 Fig), likewise, all the proteins from mammalian species were found clustered as a different group (S3 Fig). Eight GATA CREs were located in the promoter of *PxmALP* gene (Fig 3A) and four of these were found as a CRE group (Fig 3B). This group contained three CREs shared between both promoters while the fourth CRE showed sequence variation. In the promoter from the susceptible DBM1Ac-S strain this variant CRE (labelled CRE1) was found downstream of the other three whereas in the promoter from the NIL-R resistant strain the variant CRE (CRE5) was located upstream (Fig 3A and 3B). To further analyze the GATA-like CREs involved in the PxGATAd-mediated moderation of *PxmALP* promoter activity, we used different truncated promoter fragments fused to luciferase to determine their transcriptional activity. The recombinant plasmids were cotransfected with the pAC5.1b-GATAd overexpression plasmid into S2 cells. This indicated that the activity of the promoter sharply declined when the promoter was shortened from -1395/-1053 to -577/-571 upstream of ATG (Fig 3C and 3D) removing the region harbouring the CRE group (Fig 3B).

To further investigate the functional GATAd CREs, a series of truncated fragments were constructed to progressively delete the GATA-like CREs. In the susceptible promoter, the results showed that the activities of those promoters with CRE1 were significantly higher than those lacking this element (Fig 3E). This indicated that PxGATAd enhanced the activity of *PxmALP* in the promoter of susceptible DBM1Ac-S strain via interacting with GATA-like CRE1 (Fig 3E). However, the activities detected in the constructs created with the promoter region from NIL-R resistant strain showed a decrease when using fragments without CRE5, suggesting that the CRE5 is important for recruiting PxGATAd protein in the promoter region of the NIL-R resistant strain (Fig 3F).

To confirm this, we generated site directed mutants in the sequence of CRE1 and CRE5. Reciprocal mutants were constructed by exchanging the nucleotides (Mu1) between genomic sequences from susceptible and resistant strains and mutants were also made to the core binding motifs (Mu2) of both CRE variants (Figs 4A, 4B and S1). Luciferase activities showed that the function of PxGATAd was abolished in mutants for both CRE1 and CRE5 (Fig 4C and 4D). These data suggested that the PxGATAd function, modulating the transcriptional activity of *PxmALP* promoter, might be induced by the binding of PxGATAd to these specific CREs and that the loss of ability to recruit PxGATAd proteins might be affected by the changes of CRE structure introduced by the mutations.

To further investigate whether PxGATAd induced the expression of *PxmALP* through binding to CRE1 or CRE5, we synthesized CRE1 and CRE5 DNA biotin labelled probes to perform electrophoretic mobility shift assays (EMSA). Initially these were performed using nucleoproteins purified from fourth-instar larvae. The results showed a specific band shift that was observed with both the CRE1 and CRE5 probes (Fig 5A and 5B). Dose-responses were observed when the number of nucleoproteins were increased, and competition with unlabelled probes demonstrated a specific interaction (Fig 5A and 5B). Additionally, a GATAd-His

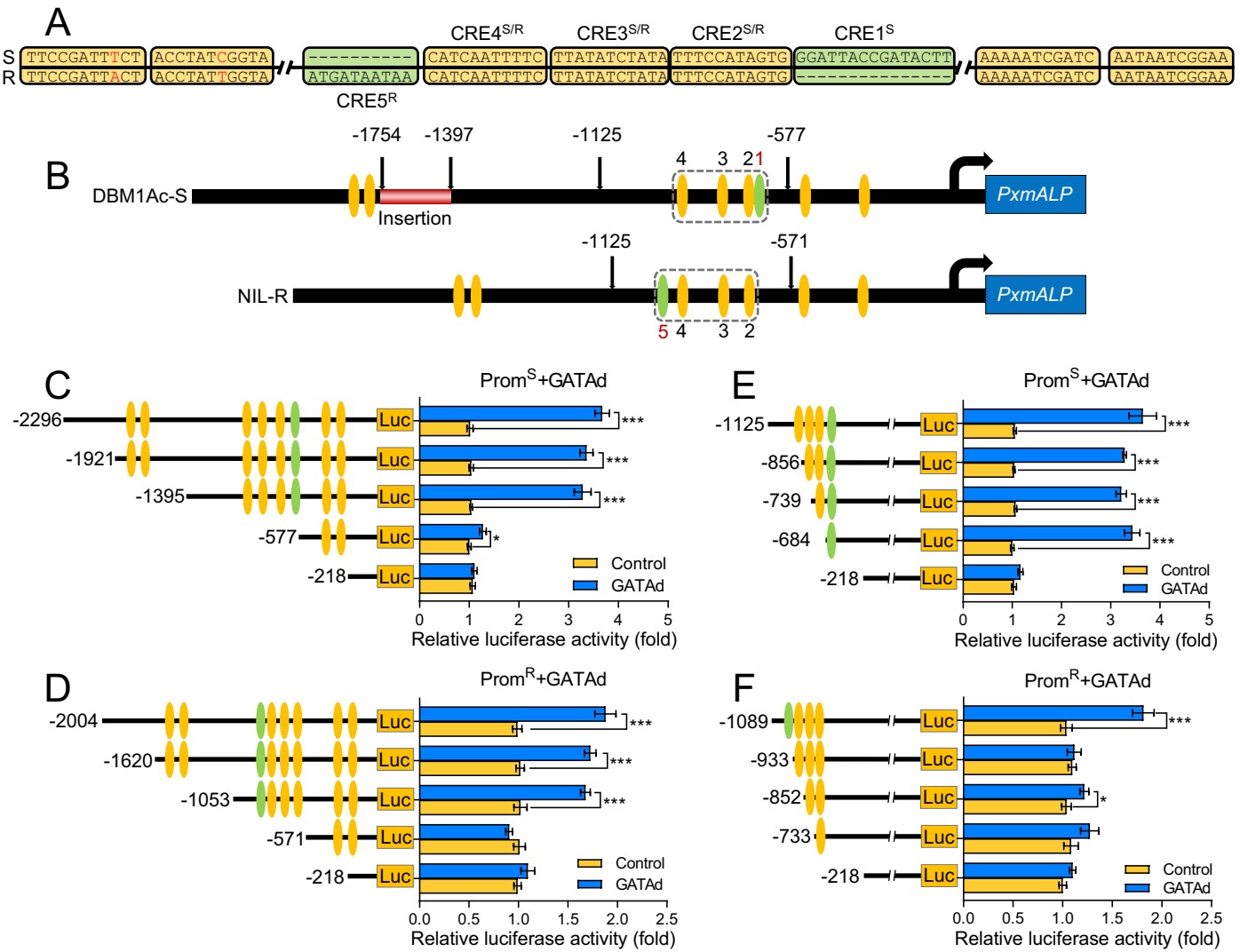

**Fig 3. PxGATAd increased the transcriptional activity of the *PxmALP* promoter via different GATA-like CREs.** (A) Sequence alignment of eight GATA-like CREs in the *PxmALP* promoter from susceptible and resistant strains. (B) Diagram of the putative GATA-like CREs in the core regulatory region of the *PxmALP* promoter. The GATA-like CREs are indicated by yellow ellipses, while the variant GATA-like CREs are indicated by green ellipses. The numbers reflect the CREs order. (C) and (D) Effects of PxGATAd overexpression on the progressive deletion constructs of the *PxmALP* promoter from the susceptible DBM1Ac-S strain (C), and *PxmALP* promoter from the resistant NIL-R strain (D), analyzed by dual-luciferase report system in S2 cells. (E) and (F) Effects of PxGATAd in the activity of *PxmALP* promoter constructions with deletions of specific GATA-like CREs in the core regulatory region of the *PxmALP* promoter from the susceptible strain (E), and the *PxmALP* promoter from the resistant strain (F), analyzed as indicated in panels (C) and (D). The left panel of (C), (D), (E) and (F) shows a schematic representation of the various reporter constructs. The right graphs in these panels show the means of relative luciferase activity ± SEM (n = 4). The significance of the difference between data sets was calculated using Holm-Sidak's test (*$p < 0.05$, ***$p < 0.001$).

tagged protein was expressed in *Escherichia coli* and the purified protein was then incubated with CRE1 or CRE5 probes to evaluate the binding of GATAd-His to these. The results indicated a band shift of similar size to that observed with the nucleoproteins, and which was also competed with unlabelled probes (Fig 5C and 5D). We further designed mutated probes according to the nucleotide mutations (Mu1) in each of the promoters and the core bind motif (Mu2) as described in Fig 4A and 4B. As expected, almost no shifted-bands were detected when GATAd-His or the nucleoproteins samples were incubated with these mutated probes

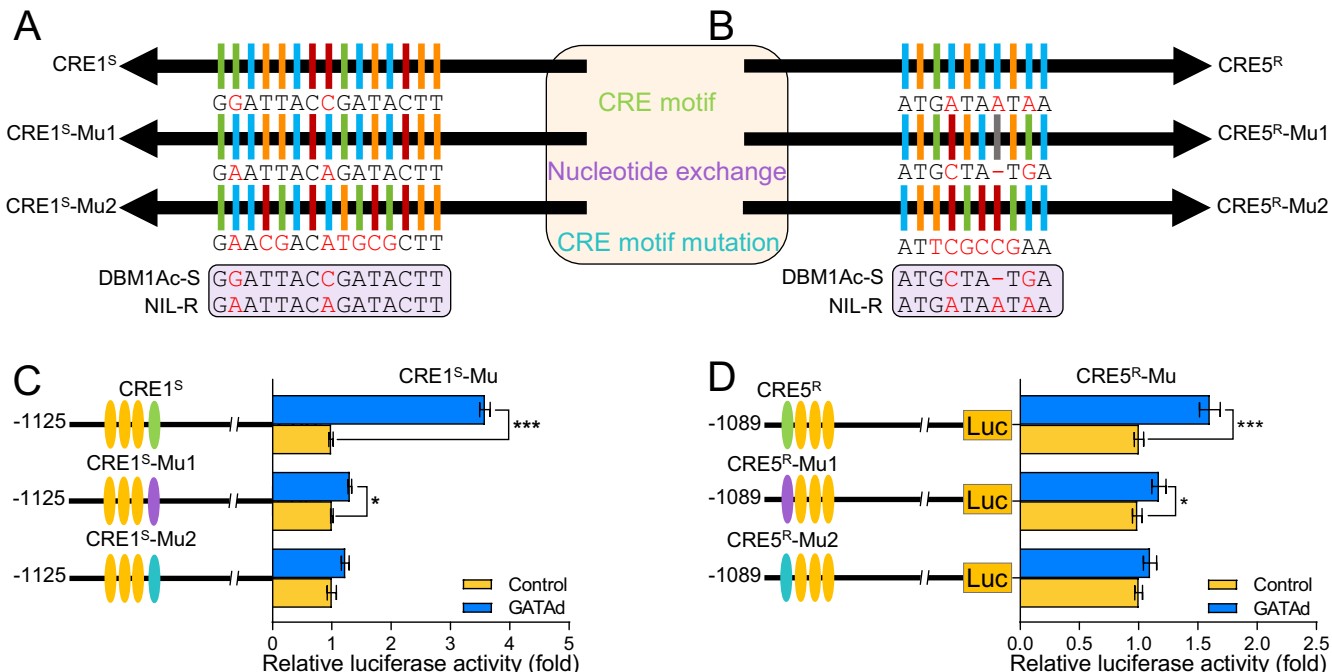

**Fig 4. Identification of the functional CRE1 and CRE5 regions.** (A) Diagram and DNA sequence of the GATA-like CRE1 and corresponding mutants. (B) Diagram and DNA sequence of the GATA-like CRE5 and corresponding mutants. (A) and (B) Different mutants generated based on the susceptible and resistant sequences are marked in the box and shown in different colours. The sequences are displayed at the bottom. The nucleotide substitutions in the mutants are highlighted by red characters. Different nucleotides are represented by different colored lines, the black arrow indicates the orientation of the CREs. (C) Effect of PxGATAd on the activity of the *PxmALP* promoter constructs containing mutations around the CRE1 motif (CRE1-Mu1) or in the core motif of CRE1 (CRE1-Mu2). (D) Effect of PxGATAd on the activity of the *PxmALP* promoter constructs containing mutations around the CRE5 motif (CRE5-Mu1) or in the core motif of CRE5 (CRE5-Mu2). The functional CREs are shown by green ellipses, the Mu1 mutants of CRE1 or CRE5 are shown in purple, the Mu2 mutants of CRE1 or CRE5 are indicated in cyan. The data shown are the means of the relative luciferase activity and their corresponding SEM values (n = 4). Holm-Sidak's test was used for statistical analysis ($^*p < 0.05$, $^{***}p < 0.001$).

(Fig 5A–5D). Subsequently, a yeast one-hybrid (Y1H) assay was performed to confirm the interaction between PxGATAd and the target or mutated (Mu2) GATA-like CREs. Overall, these results indicated that PxGATAd interacts with the target CREs directly (Fig 5E), suggesting that PxGATAd enhances the transcriptional activity of *PxmALP* by binding to CRE1 in the susceptible strain and to CRE5 in the resistant strain.

## A *cis*-acting mutation affects the regulatory performance of PxGATAd in the Cry1Ac resistant NIL-R strain

Fig 4C and 4D shows that the regulatory ability of PxGATAd to activate the *PxmALP* promoter from the NIL-R strain was significantly lower compared to the *PxmALP* promoter from the DBM1Ac-S strain. Although this could be explained by the configuration of the CRE group, we also tested the possible effect of additional sequence differences between the two promoters. Analysis of the sequence downstream of CRE5 revealed a single nucleotide substitution (C-A) and a contiguous six nucleotide insertion, in the *PxmALP* promoter from the NIL-R strain compared to DBM1Ac-S (Figs 6A and S1). To further characterize these two alterations, single mutants (Mu3 and Mu4) or a joint mutant (Mu3/4) were generated and recombined to the luciferase reporter gene (Fig 6A). Transcriptional activity of these constructions transfected into S2 cells demonstrated that the Mu3 mutant did not affect promoter activity (Fig 6B). However, the Mu4 mutant led to a significant increase in transcriptional activity. Thus, the

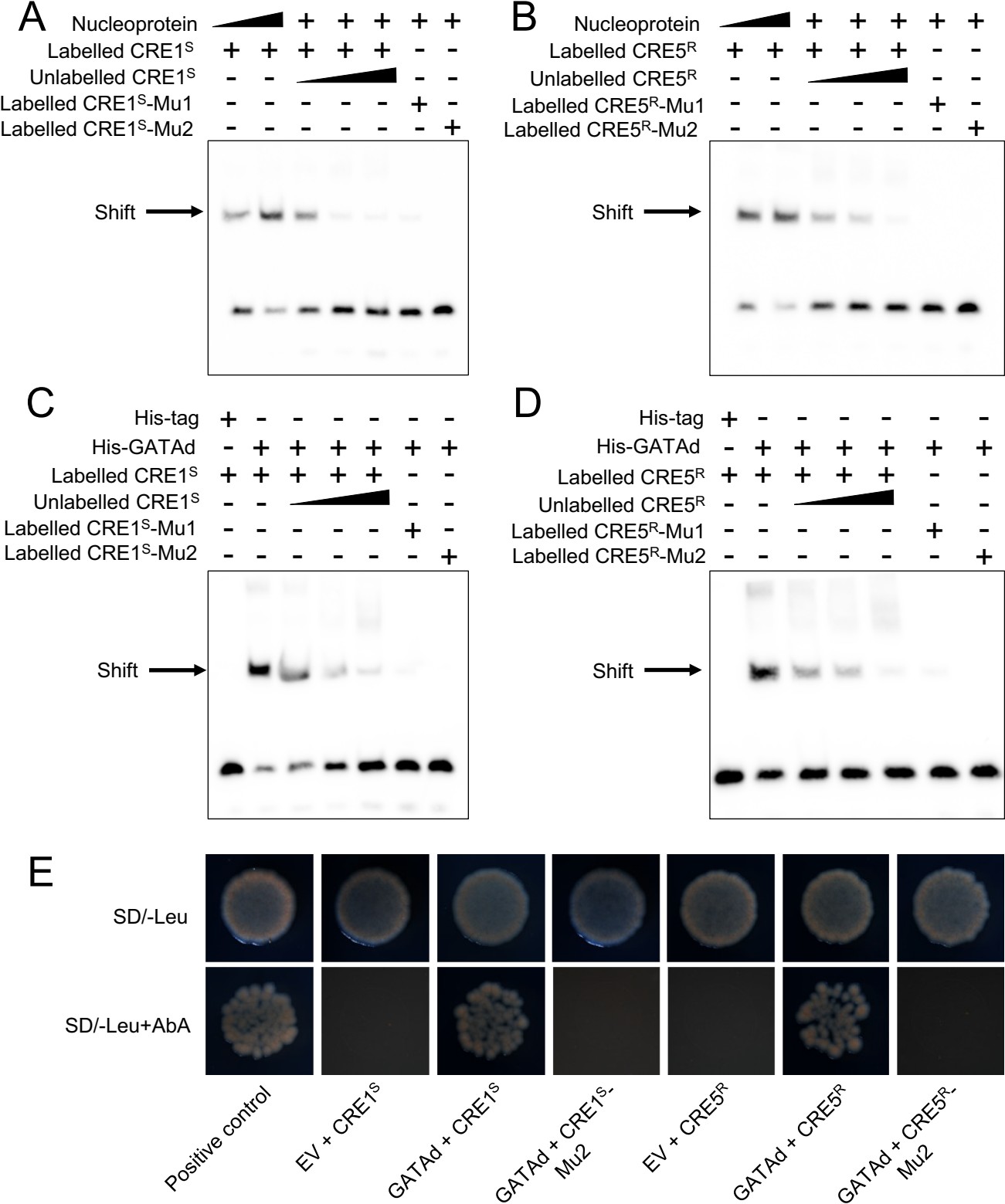

**Fig 5. Binding analysis of PxGATAd transcription factor to the GATA-like CRE1 region of *PxmALP* promoter from susceptible strain and the CRE5 region of *PxmALP* promoter from the resistant strain.** (A) and (B) Binding analysis of nucleoproteins purified from four-instar larvae to GATA-like CRE1 (A) and CRE5 (B) by electrophoretic mobility shift assays. (C) and (D) Electrophoretic mobility shift assays showing binding analysis of GATA-like CRE1 (C)

and CRE5 (D) probes to recombinant his-fused PxGATAd protein. The concentrations of GATA-like CRE1 and CRE5 probe or mutated probes were 20 fmol; the concentrations of competing probes were 100, 500, and 1000 fmol. (E) Binding assays of PxGATAd to CRE1 and CRE5 using yeast one-hybrid assay. EV, empty prey vector; positive control, transformants of pGADT7-p53 and pABAi-p53.

six-nucleotide insertion significantly decreased the transcriptional activity in the resistant *PxmALP* promoter.

## Knockdown of *PxGATAd* expression decreases *PxmALP* expression and negatively affected susceptibility to Cry1Ac toxin

To further analyze the role of *PxGATAd* in the regulation of *PxmALP*, a dsRNA targeting *PxGATAd* was synthesized. The silencing of *PxGATAd* expression by RNAi in early third-instar larvae from the DBM1Ac-S strain was performed, and we analyzed the transcript level of *PxGATAd* after 48 h in the midguts by qPCR. The results showed that *PxGATAd* mRNA levels were significantly decreased after the injection of dsPxGATAd (Fig 7A). In contrast, no alteration was observed in either of the control (buffer or dsEGFP) groups (Fig 7A). The expression of *PxmALP* was significantly reduced in *PxGATAd* silenced larvae, an observation consistent with *PxmALP* expression being under the control of *PxGATAd in vivo* (Fig 7A).

Toxicity assays at 48 h post-injection of dsPxGATAd in silenced larvae showed that reduction of *PxGATAd* gene expression decreased Cry1Ac-induced larval mortality compared to controls (Fig 7B). Specifically, 47% to 51% mortality was observed in control larvae when exposed to 1.0 mg/L of Cry1Ac protoxin, whereas only 34% mortality was counted in larvae microinjected with dsPxGATAd. When larvae were treated with 2.0 mg/L toxin, more than 91% larvae in the control samples died within three days, while the mortality declined to only 63% in those larvae injected with dsPxGATAd (Fig 7B). In the parental resistant strain, the down-regulation of multiple receptors, including *PxmALP*, has been linked to Cry1Ac resistance [36, 37, 43]. While PxGATAd clearly affects the expression of *PxmALP* it has little or no effect on these other receptors (S4 Fig), suggesting that PxGATAd is not a universal transcription factor for all Cry1Ac receptors.

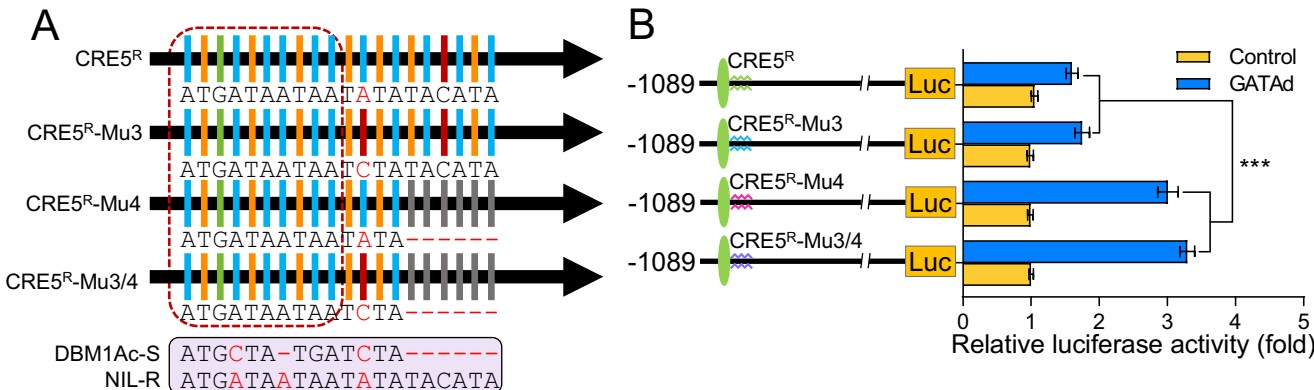

**Fig 6. Analysis of the joint effect on *PxmALP* expression of two *cis*-acting mutations in the 3′-flanking region of CRE5.** (A) Diagram and DNA sequence of 3′-flanking region from the GATA-like CRE5 and corresponding mutants. The nucleotide substitutions based on susceptible and resistant sequences in the mutants are highlighted with red characters. The sequences are displayed at the bottom. Different nucleotides are represented by different colored lines. (B) Effect of PxGATAd on the transcriptional activity of the *PxmALP* promoters containing different *cis*-acting mutations. The left panel shows a schematic representation of the various reporter constructions, CRE5 is indicated by green ellipses, the wild type of 3′-flanking sequence of CRE5 is indicated by a green wavy line, the blue and red wavy lines indicate CRE5-Mu3 and CRE5-Mu4, respectively, the joint mutant CRE5-Mu3/4 is indicated by a purple wavy line. The graph shows the means of relative luciferase activity ± SEM (n = 4). The significance of the difference between values sets was calculated using Holm-Sidak's test (***$p < 0.001$).

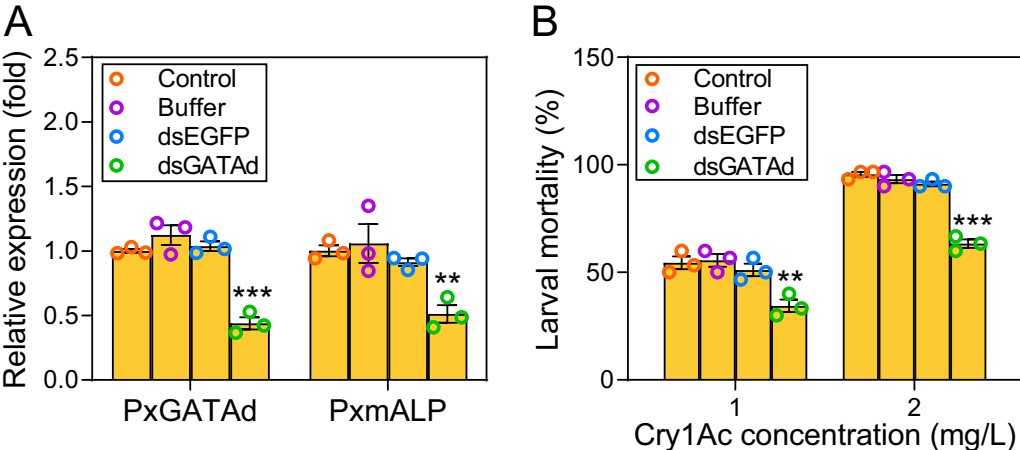

**Fig 7. Effect of *PxGATAd* gene silencing on *PxmALP* expression *in vivo*.** (A) Relative expression of *PxGATAd* and *PxmALP* in susceptible larvae after 48 h post injection with buffer, dsEGFP, or dsPxGATAd. The *RPL32* gene was used as an internal control. Untreated larvae were used as negative control, the expression level (fold) was normalized based on the value of the expression level in DBM1Ac-S strain. (B) Susceptibility of DBM1Ac-S larvae to two concentrations of Cry1Ac protoxins after injection with buffer, dsEGFP or dsPxGATAd. Untreated larvae were used as control. (A) and (B) The significance of the difference between values sets was calculated using Holm-Sidak's test (**$p < 0.01$, ***$p < 0.001$, n = 3).

## MAPK signaling negatively regulates *PxGATAd* transcriptional levels

Previous studies demonstrated that *PxmALP* is involved in Bt resistance and that it is down-regulated under the guidance of the MAPK signalling pathway in the NIL-R strain. *PxMAP4K4*, a MAPK upstream gene, is constitutively overexpressed in the Bt resistant NIL-R

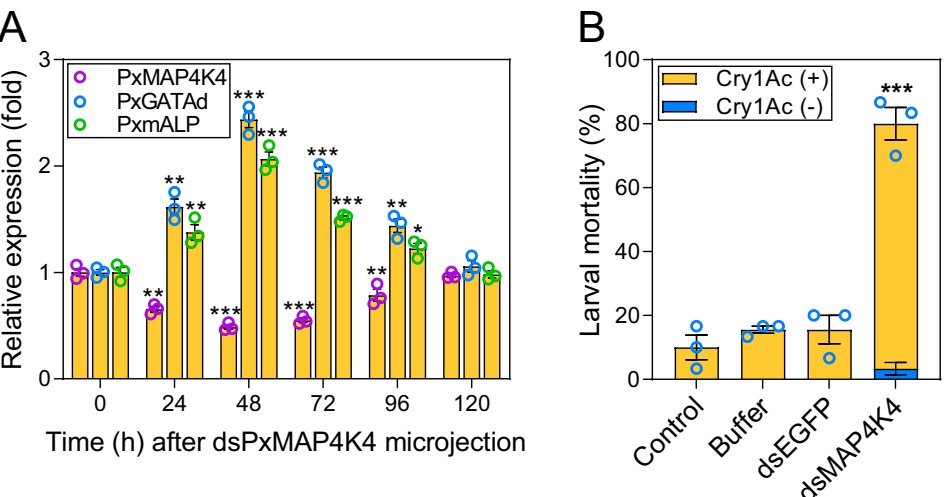

**Fig 8. Repressing *PxMAP4K4* gene restores the expression of *PxGATAd* and *PxmALP* and susceptibility to Cry1Ac toxin.** (A) Effect of *PxMAP4K4* gene silencing on *PxGATAd* and *PxmALP* expression. The expression of *PxMAP4K4*, *PxGATAd* and *PxmALP* was analyzed in the midgut tissue from larvae of NIL-R strain after different time periods post-injection with dsPxMAP4K4. The expression level (fold) of genes was normalized based on the value of the expression level detected at 0 h. (B) Susceptibility to Cry1Ac protoxin (LC$_{10}$, 1000 mg/mL) in resistant NIL-R larvae injected with buffer, dsEGFP or dsPxMAP4K4. The mortality of untreated larvae was used as control. (A) and (B) The data are presented as mean values ± the SEM, the significance of the difference between values sets was calculated using Holm-Sidak's test (*$p < 0.05$, **$p < 0.01$, ***$p < 0.001$, n = 3).

strain and *trans*-regulates the expression of *PxmALP* [36]. These data led us to determine whether the expression of *PxGATAd* is controlled by MAPK signaling. To determine the possible relationship between *PxMAP4K4* and *PxGATAd* expression, *PxMAP4K4* expression was silenced in NIL-R larvae by microinjection of dsPxMAP4K4 RNA. The transcript levels of *PxMAP4K4*, *PxGATAd* and *PxmALP* were determined at different times and showed that the expression of *PxMAP4K4* was significantly repressed at 24–96 h post injection (Fig 8A). We also found that the transcript levels of *PxGATAd* and *PxmALP* were enhanced up to 2.4- and 2.0-fold, respectively at 48 h post-injection (Fig 8A). As previously observed, the bioassay results indicated that the injected larvae dramatically restored susceptibility to toxin (Fig 8B). Overall, these data indicated that *PxGATAd* serves as a responsive factor to MAPK signaling, and that the activated MAPK cascade represses *PxGATAd* expression, resulting in reduced expression of *PxmALP* and contributing to Cry1Ac resistance in *P. xylostella*.

## Discussion

Here, we demonstrate that the transcription factor PxGATAd serves as a major mediator of the MAPK signaling pathway in the expression of the larval midgut protein PxmALP. This enzyme is a known receptor for the Cry1Ac toxin and its down-regulation results in decreased susceptibility to this toxin [36]. Both *trans*- and *cis*-acting factors have been found to contribute to this down-regulation as summarized in Fig 9. However, the knockdown of *PxGATAd* in the DBM1Ac-S strain, which doesn't have the CRE5 variant element, resulted in a circa 50% drop of both *PxGATAd* and *PxmALP* (Fig 7). This is not too dissimilar to the situation in the resistant NIL-R strain in which decreases of 50% in *PxGATAd* (Fig 2) and 60% in *PxmALP* [36] were observed, indicating that the *trans*-effect is the dominant control measure. Although

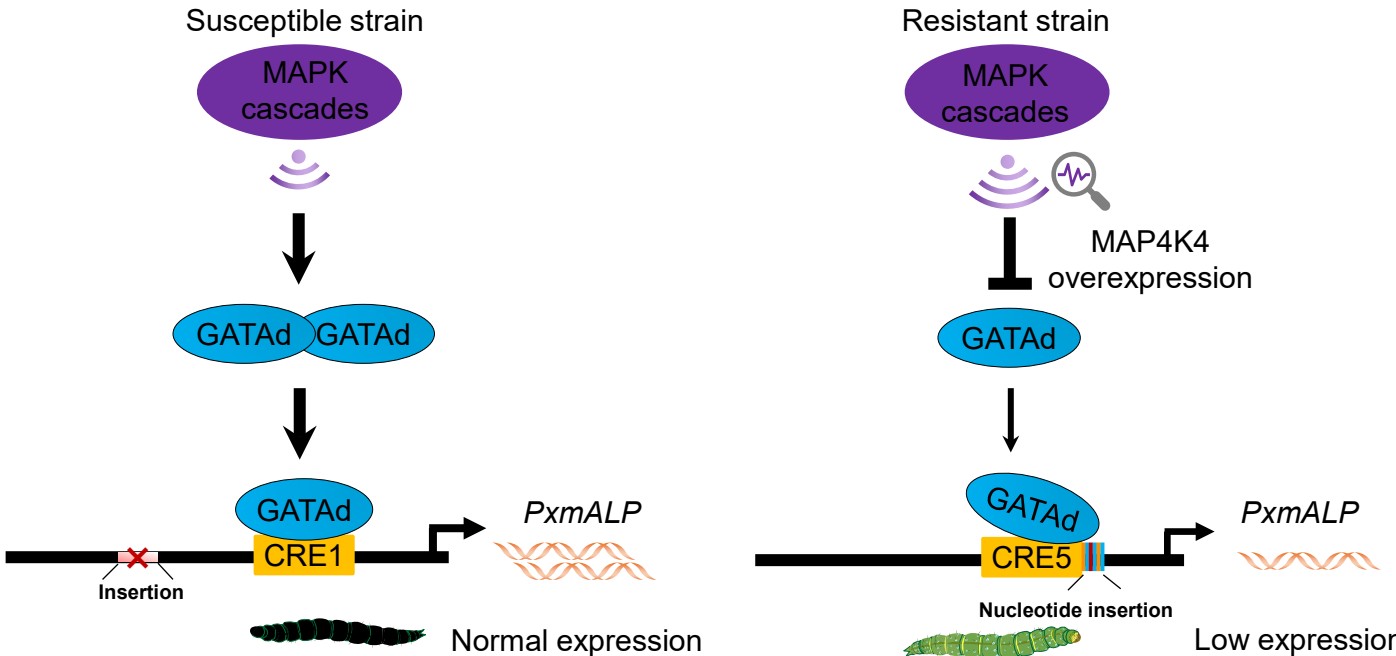

**Fig 9. Diagram showing a model of the PxGATAd mechanism involved in the regulation of *PxmALP* expression.** Transcription factor PxGATAd serves as a positive regulatory factor by binding with different GATA-like CREs to regulate transcription from the *PxmALP* promoter region from Cry1Ac susceptible and resistant strains, PxGATAd is *trans*-regulated by the upstream activated MAPK signaling pathway. In the Cry1Ac susceptible strain, a large fragment insertion (represented by a red box) was detected. In the resistant strain, a *cis*-acting element mutation generated by a six-nucleotide insertion near the CRE5 element negatively affects the binding of PxGATAd to CRE5, repressing the transcription of *PxmALP*. Therefore, both *cis*-acting element mutation and *trans*-regulatory factors are involved in regulating transcription of *PxmALP*, thus mediating Cry1Ac resistance in *P. xylostella*.

PxGATAd is involved in the MAP4K4-dependent expression of *PxmALP*, it does not appear to be involved in the reduced expression of the other receptors, a fact that is consistent with the observation that the reduced susceptibility to Cry1Ac observed in the *PxGATAd* knock down experiment is significantly less than that observed in the parental strain.

Different TFs are involved in multiple functions such as organism physiology, disease, and phenotypic diversity [44], establishing cell type-specific gene expression programs and developmental patterns [45]. More often, they control specific signaling pathways such as those involved in immune responses [46]. GATA zinc-finger TFs play crucial roles in development and differentiation. Various GATA proteins are expressed in eukaryotes and show multiple functions. In vertebrates, they are critical for hematopoiesis and mesoendodermal development [47, 48]. In invertebrates, GATA proteins are involved in multiple functions, for example, in *Drosophila* larvae, the GATA factor *Serpent* (*srp*, GATAb), plays a vital role in the tissue-specific expression of immunity genes, by promoting a systemic induction of antimicrobial peptides in response to infection [49]. As in vertebrates and *Drosophila*, the role of endodermal GATA TFs in regulating epithelial innate immune responses is conserved in *Caenorhabditis elegans* [50]. Recently, GATAe was shown to be involved in the transcriptional activation of *H. armigera* Cry1Ac toxin receptor genes, including *HamALP* [51]. In the case of *H. armigera*, the role of GATAd *in vitro* or *in vivo* was not analyzed since it showed reduced midgut expression [51]. Here, we show that the homologous TF GATAd plays a crucial role in mediating Cry1Ac resistance via regulating the differential expression of the midgut receptor, *PxmALP*, in *P. xylostella*. GATAd and GATAe are orthologous proteins that may have redundant functions [52]. Overall, our results are the first to establish the function of GATAd in mediating Bt toxin resistance *in vivo* in *P. xylostella*.

TFs vary in how they influence transcription via DNA binding specificities [53]. They can recruit RNA polymerase directly, or alternatively can recruit other TF cofactors that lead to activation or repression of gene expression [44]. The homo- and hetero-dimer mediated activation of transcription is regarded as a crucial mechanism to modify TF activity and function. For instance, the homeodomain proteins POU and Abd-A are well known to regulate insect development and metamorphosis [19]. The heterodimer bZIP TFs, CncC/Maf, play predominate roles in regulating responses associated with pathogen defence and abiotic stresses responses in plants [54, 55]. TFs are also involved in the regulation of the overexpression of detoxification genes associated with metabolism of xenobiotics, including some insecticides, in insects [56, 57]. GATA TFs, along with FoxA, are considered as pioneer TFs that bind to condensed chromatin allowing the interaction of additional TFs that activate transcription [58, 59]. For example, two related zinc finger proteins, USH (U-shaped) and FOG (Friend of GATA-1) have been described as cofactors of GATA proteins, Pannier and GATA-1, respectively, modulating their transcriptional activities in mammals [60]. However, it still remains to be studied if there are any other TFs that assist the PxGATAd to exert its regulatory function in *P. xylostella* and other insects.

TFs can act on multiple target genes to balance the growth, development and environmental adaptation of living organisms [61]. It has been reported that FoxA is involved in the transcriptional activation of *Spodoptera litura* ABCC2/3 transporters, which serve as Cry1Ac receptors [62], but our results show that FoxA does not participate in the expression of *PxmALP* in *P. xylostella*. GATAe has been identified in cell lines as being able to affect the expression of Cry toxin receptors [51]. Our results found that the related *PxGATAd* showed slight regulatory activity towards *APN3a* and *ABCC2* but not for for *APN1*, *ABCB1*, *ABCC3* and *ABCG1* (S4 Fig). Recently, TF Jun activated by the MAPK signaling pathway has been demonstrated to paticipate in Cry1Ac resistance by repressing the expression of *ABCB1* in *P. xylostella* [63].

It has been reported that *cis*-regulatory mutations could result in *trans*-regulatory differences affecting gene expression [14]. There are substantial variations in gene expression between individuals, populations and species. Generally, the evolution of gene regulation and its expression is considered to contribute to adaptation [17]. In insects, *cis*-regulatory changes in the promoter of genes involved in insecticide resistance adjusted the transcriptional output through copy number variation to defend against pesticide action [64, 65]. The significance of *cis*-acting mutations on detoxification enzyme genes for pesticides resistance has been widely recognized [22, 66, 67]. Recently, it was shown that a *cis*-mutation in the promoter of *ABCG1* reduced the promoter activity by affecting the binding of the TF Antennapedia (Antp), resulting in Cry1Ac resistance in *P. xylostella* [33]. Here, we further identified *cis*-acting mutations in the promoter of *PxmALP* that influenced expression of this gene. The GATA CRE in the susceptible strain recruits the PxGATAd protein to regulate the normal expression of *PxmALP* gene, the mutations occurring in the resistant strain resulted in the loss of this original CRE (Fig 3). Although PxGATAd can interact with a variant CRE in the resistant strain its effect is much weaker than in the susceptible. The down-regulation of Cry1Ac receptors is a dynamic response to intoxication that has become constitutive in the resistant population [36, 37, 43]. Since *PxmALP* expression is reduced in the resistant population, there is less selective pressure to retain normal expression levels of this enzyme and so mutations that affect this can be tolerated. If such mutations further enhance the resistance phenotype (as is the case here) then they themselves will be retained while the strain is under continued selection pressure.

Generally, MAPK pathways are triggered in response to various extracellular stimuli, tuning the function of multiple transcription activators. MAPK directed activation of CREB resulted in imidacloprid resistance in the whitefly *Bemisia tabaci* by elevating the expression of *CYP6CM1* [68]. In addition, it has been shown that MAPKs mediate GATA-2 nuclear translocation in the signal cascade transducing LPS signal resulting in induced *il-1β* gene expression in macrophages [69]. Finally, activation of MAPK p38 kinase is indispensable for hypertrophic agonist-induced GATA-4 binding to B-type natriuretic peptide gene and sufficient for GATA-dependent B-type natriuretic peptide gene expression [70]. Here, we show that down-regulation of *PxMAP4K4* expression resulted in increased expression of *PxGATAd*, indicating that *PxGATAd* was a responsive factor to MAPK signaling pathways.

In summary, this work extends our previous findings by identifying a key responsive factor *PxGATAd* of the *PxMAP4K4* signal transduction pathway involved in delivering Bt Cry1Ac toxin resistance. Our study helps unravel the complex response mechanism that the insect has developed to defend against a bacterial pathogen.

## Materials and methods

### Insect rearing

Cry1Ac toxin-susceptible *P. xylostella* DBM1Ac-S strain and its near-isogenic Cry1Ac-resistant NIL-R strain were used in the study [71]. The DBM1Ac-S strain was not exposed to any Bt products or to Cry1Ac toxin. The NIL-R strain has been continuously exposed to Cry1Ac protoxin to maintain the high resistance level. It exhibits over 4000-fold resistance to Cry1Ac protoxin to the susceptible strain. All larvae were reared on fresh cabbage leaves and maintained in our insectary under a photoperiod of 16L:8D and 65% RH at 25°C, while all adults were supplied with 10% honey/water solution.

### Toxin preparation and toxicity bioassay

The Cry1Ac protoxin was prepared from Bt var. *kurstaki* strain HD-73 and quantified using Bradford's method with BSA as a standard [72]. Protoxin was solubilized in 50 mM $Na_2CO_3$

(pH 9.6) and stored at -20˚C before used. 72 hours leaf-dip bioassays were implemented to assess toxicity of Cry1Ac protoxin to *P. xylostella* larvae as described previously [73, 74]. Thirty third-instar larvae were selected from normal strains or dsRNA injected strains and tested with different toxin concentrations (1 mg/mL and 2 mg/mL), each bioassay was replicated three times.

## RNA isolation, cDNA synthesis and genome DNA extraction

Total RNA was extracted from fourth-instar larvae and midgut tissues collected from susceptible or resistant strains using TRIzol reagent (Invitrogen) according to manufacturer's protocols. cDNA was reverse transcribed using the PrimeScript II 1st strand cDNA Synthesis Kit (TaKaRa) for gene cloning, or using the PrimeScript RT kit (containing gDNA Eraser, Perfect Real Time) (TaKaRa) for qPCR analysis using 1 μg of total RNA and following the instruction manual. Genomic DNA (gDNA) samples were isolated from DBM1Ac-S and NIL-R individuals using a TIANamp Genomic DNA Kit (TIANGEN) according to the manufacturer's instructions. Integrity of the nucleic acid for each sample was analyzed in 1% TBE agarose gel electrophoresis, and the concentration was determined by using a NanoDrop 2000c spectrophotometer (Thermo Fisher Scientific Inc.). All samples were kept at -20˚C before use.

## Cloning of the *PxmALP* 5′-flanking regions and dual luciferase activity analysis

To obtain the promoter sequence of *PxmALP*, gDNA extracted from single 4th instar larva was prepared. Primers for cloning the promoter (S1 Table) were designed using Primer3Plus (http://www.primer3plus.com/cgi-bin/dev/primer3plus.cgi?tdsourcetag=s_pcqq_aiomsg) according to the genome sequence downloaded from the diamondback moth genome database (DBM-DB) (http://116.62.11.144/DBM/index.php) and the lepidopteran genome database Lepbase (http://ensembl.lepbase.org/Plutella_xylostella_dbmfjv1x1/Info/Index). The target sequences were amplified with PrimerSTAR Max DNA Polymerase (Takara), the fragments were then ligated to the *pEASY*-Blunt Cloning vectors (TransGen Biotech) and confirmed by DNA sequencing. Promoter region sequences were amplified and added to the pGL4.10 vector (Promega), and various truncated promoter fragments of *PxmALP* were obtained using the full-length promoter plasmid as template and specific primers (S1 Table). Mutated promoter plasmids were generated by a M5 Site-Directed Mutagenesis Kit (Mei5 Biotechnology) following the instruction manual and specific primers (S2 Table). The putative transcription factor binding sites (TFBS) were predicted by ALLGEN (http://alggen.lsi.upc.es/cgi-bin/promo_v3/promo/promoinit.cgi?dirDB=TF_8.3) and JASPAR (http://jaspar.genereg.net/) software with the "Insecta" group *in silico*. The over expression constructs of each TF were produced by amplifying the CDS region into the pAC5.1b vector using specific primers (S3 and S4 Tables).

S2 cells were cultured in 24-well plates and kept at 27˚C. The promoter constructs were transfected or co-transfected with expression plasmids into S2 cells when the cells density approached $5×10^5$. 600 ng promoter constructs with full-length, truncated or mutated promoter, and 200 ng reference reporter pGL4.73 plasmid were co-transfected to the cells with the assistance of Lipofectamine 2000 transfection reagent (Invitrogen). For detecting the trans-activation of TFs, 200 ng of various promoter constructs and 100 ng pGL4.73 vector were co-transfected with 600 ng expression plasmids. Finally, luciferase activity was determined by a Dual-Luciferase Report Assay System (Promega) at 48 h post-transfection. The luciferase activity was normalized to the Renilla luciferase activity.

## Electrophoretic mobility-shift assay (EMSA)

EMSAs were used to analyze PxGATAd binding to the putative GATA CREs in the promoter of *PxmALP* as follows. Firstly, probes containing the putative binding site motifs or mutated sequences were labelled with biotin at the 5′-end (synthesized by TSINGKE, Tianjin, China). Total nuclear proteins were extracted from fourth-instar larvae using a Nuclear and Cytoplasmic Protein Extraction Kit (Beyotime Biotechnology, China). Recombinant PxGATAd was expressed with His-tag in *Escherichia coli* and purified with a His-tag Protein Purification Kit (Beyotime Biotechnology, China). EMSAs were performed using a LightShift Chemiluminescent EMSA Kit (Thermo Scientific) according to the manufacturer's instructions. Briefly, the DNA-protein binding mixtures were incubated for 20 min at room temperature, then loaded into 6% (w/v) native polyacrylamide gels and electrophoresed in 0.5× TBE buffer (Solarbio, China). After electrophoresis, the DNA-protein complexes were transferred onto a positively charged nylon membrane (Beyotime Biotechnology, China). The membrane was then cross linked under the UV lamp and finally incubated with stabilized streptavidin-horseradish peroxidase conjugate for chemiluminescence detection (Thermo Scientific). The chemiluminescence signals were captured by the Tanon-5200 Chemiluminescent Imaging System (Tanon).

## Yeast one-hybrid assay

Yeast one-hybrid (Y1H) assay was performed as follows. Three tandem copies of GATA-like CREs (CRE1/5′-TGT <u>GGA TTA CCG ATA CTT</u> GTC-3′, CRE5/5′-ACC <u>ATG ATA ATA ATA TAT</u> AC-3′) and mutated GATA-like CREs (CRE1-M/5′-TGT <u>GAA CGA CAT GCG CTT</u> GTC-3′, CRE5-M/5′-ACC <u>ATT CGC CGA A</u>TA TAT AC-3′) were inserted into the pAbAi vector between the *Xho*I and *Hind*III restriction sites to generate bait plasmids. The recombinant bait plasmid was then linearized with *Bst*BI and finally integrated into Y1HGold yeast to construct the bait strains. The bait strains were grown on SD/-Ura medium under the selection of 500 ng/mL aureobasidin A (AbA). The PxGATAd full length coding sequence was amplified with specific primers (S4 Table) and inserted into pGADT7 vector to generate the prey plasmid AD-GATAd and then transformed into the bait yeast strain carrying GATAd-like CREs or mutated elements that were analyzed separately. The Y1HGold strain cotransformed with the pGADT7-p53 and pAbAi-p53 plasmids was used as positive control, and the negative control was the Y1HGold strain cotransformed with the empty vector pGADT7 and the normal pAbAi-CRE plasmid. Transformants were grown on SD/-Leu medium with 500 ng/mL AbA according to the instructions Matchmaker Gold Yeast One-Hybrid System (Clontech).

## qPCR analysis

The cDNA reversed from midgut total RNA was applied for qPCR analysis. The qPCR experiments were performed using a QuantStudio 3 Real-Time PCR System (Applied Biosystems, USA) using FastFire qPCR PreMix (SYBR Green) (Tiangen, Beijing, China) according to the manufacturer's protocol. A typical two-step protocol was adopted. Briefly, initial denaturation at 95˚C for 1 min, followed by 40 cycles of 95˚C for 5 s, 60˚C for 15 s. The relative expression levels were calculated using the $2^{-\Delta\Delta Ct}$ method, with ribosomal protein *L32* (*RPL32*) gene (GenBank accession no. AB180441) as the reference control. Melting curve analysis was performed to confirm the specificity of amplification. Each experiment was conducted with three biological replicates. One-way ANOVA with Holm-Sidak's test were used to evaluate the significant statistical differences between each treatment. All primers used in qRT-PCR are listed in S5 Table.

## RNA interference

Synthesis of dsRNA and microinjection were carried out as described elsewhere [75]. In brief, an optimum target position of *PxGATAd* and *PxMAP4K4* was selected to synthesize dsRNA. Gene-specific dsRNA primers containing T7 promoter sequence on the 5′-end were designed (S5 Table). The dsRNAs were generated using the T7 Ribomax Express RNAi System (Promega), the resulting dsRNAs were stored at -80˚C. dsRNA of EGFP fragments were also generated as negative controls. Subsequently, the synthesized dsRNAs were suspended in injection buffer (10 mM Tris-HCl, pH 7.0; 1 mM EDTA) and blended with Metafectene PRO transfection reagent (Biontex) in a 1:1 volume ratio, the mixture was incubated for 20 min at room temperature. The microinjection was conducted to the hemocoel of newly molted third-instar larvae, dsEGFP (300 ng) or dsRNA (300 ng) by using a Nanoliter 2000 microinjection system (World Precision Instruments). More than thirty larvae were injected for each treatment and three independent experiments were conducted. The injected larvae were transferred onto the fresh cabbage leaves under normal rearing culture condition for qPCR assays to monitor the silencing efficiency and used in bioassays. One-way ANOVAs with Holm-Sidak's test was used to evaluate statistically significant differences between each group for qPCR analysis of the relative expression and for bioassay analysis.

## Supporting information

**S1 Fig. Sequence alignment of the 5′-flanking regions of *PxmALP* cloned from Cry1Ac susceptible DBM1Ac-S and resistant NIL-R strains.** The nucleotides are numbered relative to the translation start site (ATG) indicated as +1 and in red font. The position also highlighted in red and with an arrow indicates the transcription start site (TSS). GATA-like CREs are highlighted inside red boxes.
(PDF)

**S2 Fig. Multiple amino acid sequence alignment of different GATAd proteins.** The conserved GATA-type zinc finger (Znf) domain of GATAd proteins is highlighted by a red background. PxGATAd (*Plutella xylostella*, MZ712004), BmGATAd (*Bombyx mori*, XP_012546211.1), TcGATAd (*Tribolium castaneum*, EFA09251.2), AaGATAd (*Aedes aegypti*, EAT41982.1), HsapGATAd (*Homo sapiens*, NP_001295023.1), PhcGATAd (*Pediculus humanus corporis*, XP_002428265.1), DmGATAd (*Drosophila melanogaster*, NP_001260326.1).
(PDF)

**S3 Fig. Phylogenetic relationship of GATAd proteins in insects and mammals.** Phylogenetic analysis of the GATAd genes was performed by maximum likelihood method. The GATAd encoding sequence was corrected using NCBI database and transcriptome data of *P. xylostella* and then cloned from fourth-instar larvae. PxGATAd is indicated by a red star. Full-length amino acid sequences of GATAd genes were retrieved from the GenBank database. Abbreviation: **1. Hymenoptera** [**Am** (*Apis mellifera*, XP_001120276.2); **Cf** (*Camponotus floridanus*, XP_025270700.1); **Bte** (*Bombus terrestris*, XP_003400693.1); **Mr** (*Megachile rotundata*, XP_003708269.1); **Nf** (*Nylanderia fulva*, XP_029159148.1); **Ae** (*Acromyrmex echinatior*, XP_011057942.1); **Hsal** (*Harpegnathos saltator*, XP_011148075.1); **Hh** (*Halyomorpha halys*, XP_014279044.1); **Cl** (*Cimex lectularius*, XP_014261884.1)]; **2. Anoplura** [**Phc** (*Pediculus humanus corporis*, XP_002428265.1]; **3. Coleoptera** [**Dpo** (*Dendroctonus ponderosae*, XP_019764087.1); **Ld** (*Leptinotarsa decemlineata*, XP_023023616.1); **Atu** (*Aethina tumida*, XP_019881566.1); **Nv** (*Nicrophorus vespilloides*, XP_017769919.1); **Apl** (*Agrilus planipennis*, XP_018334804.1); **Tc** (*Tribolium castaneum*, EFA09251.2); **Ot** (*Onthophagus taurus*, XP_022900328.1); **Ob** (*Oryctes borbonicus*, KRT80118.1)]; **4. Lepidoptera** [**Dpl** (*Danaus*

*plexippus*, OWR52456.1); **Pr** (*Pieris rapae*, XP_022123638.1); **Pxy** (*Plutella xylostella*, MZ712004); **Atr** (*Amyelois transitella*, XP_013199688.1); **Pm** (*Papilio machaon*, XP_014367479.1); **Pxu** (*Papilio xuthus*, XP_013169273.1); **Bm** (*Bombyx mori*, XP_012546211.1); **Ha** (*Helicoverpa armigera*, XP_021183187.1); **Sl** (*Spodoptera litura*, XP_022834873.1); **Tn** (*Trichoplusia ni*, XP_026725121.1)]; **5. Diptera** [**Aa** (*Aedes aegypti*, EAT41982.1); **Cq** (*Culex quinquefasciatus*, EDS42992.1); **Dm** (*Drosophila melanogaster*, NP_001260326.1); **Cc** (*Ceratitis capitata*, XP_004537903.1); **Zc** (*Zeugodacus cucurbitae*, XP_011186611.2)]; **6. Mammalia** [**Ss** (*Sus scrofa*, NP_999458.1); **Fc** (*Felis catus*, XP_011279866.2); **Bta** (*Bos taurus*, CAC69835.1); **Oc** (*Oryctolagus cuniculus*, XP_008247101.1); **Clu** (*Canis lupus*, XP_025318689.1); **Mm** (*Mus musculus*, BAE22221.1); **Hsap** (*Homo sapiens*, NP_001295023.1); **Rn** (*Rattus norvegicus*, XP_006252251.1)].
(PDF)

**S4 Fig. Role of PxGATAd on the transcriptional regulation of other Cry1Ac-receptors.** The data are represented by the means and the corresponding SEM values. Four independent transfections were conducted for each pair of plasmids. Holm-Sidak's test was used for statistical analysis ($^{*}p < 0.05$, $^{***}p < 0.001$).
(PDF)

**S1 Table. Sequence of the primers used for cloning the promoter region of receptor genes and construction of pGL4.10 recombinant plasmids.**
(PDF)

**S2 Table. Sequence of the primers used for site-directed mutagenesis.**
(PDF)

**S3 Table. Sequence of the primers used for cloning TFs from *P. xylostella*.**
(PDF)

**S4 Table. Primers used for constructing the recombinant plasmids of TFs.**
(PDF)

**S5 Table. Sequence of the primers used for real time qPCR analyzes and dsRNA synthesis.**
(PDF)

**S1 Data. Raw data used in the figures and statistical analysis.**
(XLSX)

## Author Contributions

**Conceptualization:** Le Guo, Neil Crickmore, Zhaojiang Guo, Youjun Zhang.

**Funding acquisition:** Zhaojiang Guo, Youjun Zhang.

**Investigation:** Le Guo, Zhouqiang Cheng, Jianying Qin, Dan Sun.

**Resources:** Shaoli Wang, Qingjun Wu.

**Supervision:** Zhaojiang Guo, Youjun Zhang.

**Writing – original draft:** Le Guo, Zhaojiang Guo.

**Writing – review & editing:** Neil Crickmore, Xuguo Zhou, Alejandra Bravo, Mario Soberón, Zhaojiang Guo, Youjun Zhang.

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
