## [Decision Letter · Decision Letter 0]

29 Dec 2021

Dear Dr Zhang,

Thank you very much for submitting your Research Article entitled 'MAPK-mediated transcription factor GATAd contributes to Cry1Ac resistance in diamondback moth by reducing PxmALP expression' to PLOS Genetics.

The manuscript was fully evaluated at the editorial level and by independent peer reviewers. The reviewers appreciated the attention to an important problem, but raised some substantial concerns about the current manuscript. Based on the reviews, we will not be able to accept this version of the manuscript, but we would be willing to review a much-revised version. We cannot, of course, promise publication at that time.

If you decide to revise the manuscript for further consideration at PLOS Genetics, please aim to resubmit within the next 60 days, unless it will take extra time to address the concerns of the reviewers, in which case we would appreciate an expected resubmission date by email to plosgenetics@plos.org.

[LINK]

We are sorry that we cannot be more positive about your manuscript at this stage. Please do not hesitate to contact us if you have any concerns or questions.

Yours sincerely,

Subba Reddy Palli, Ph.D.

Associate Editor

PLOS Genetics

Gregory P. Copenhaver

Editor-in-Chief

PLOS Genetics

Reviewer's Responses to Questions

**Comments to the Authors:**

Reviewer #1: In this manuscript the genetic mechanisms behind down-regulation of PxmALP in a Cry1Ac resistant strain of Plutella xylostella are characterized. Both a transcription factor and a cis-acting sequence are identified as participating in down-regulation of PxmALP in the resistant insects. The data presented advances our understanding of the genetic control of Cry toxin receptors and how insects may down-regulated their expression to reduce susceptibility to these toxins.

-The manuscript is well written, although I consider the choice of references is not appropriate in some instances:

Line 73: This sentence is about evolution of resistance to pesticides worldwide, yet reference 1 is too specific to Drosophila melanogaster, which is not a relevant pest of crops. A reference to a review on insect resistance to pesticides would be more appropriate.

Line 75: This statement is on Bt as alternative to synthetic pesticides, yet reference 2 refers to mechanisms of resistance to Bt and does not provide any details on comparisons to synthetic pesticides. A review on the use and benefits of Bt versus chemical pesticides would be more appropriate.

Line 86: Reference 12 is the same as reference 3 and is focused on mechanisms of resistance, not on describing Bt midgut receptors. Reference 35 would be more appropriate here.

Line 88: Reference 12 (or 3) would be more appropriate here as it presents a review of cases in which reduced expression of receptors are involved in resistance to Bt toxins.

Line 122: Reference 12 does not provide any data supporting the physiological role of ALPs. Probably no reference is needed here as this is well established.

-Fig. 2 shows that GATAd expression is 50% reduced in resistant compared to susceptible larvae, yet in Fig. 7 it is shown that this same level of knock down results in a slight difference (albeit significant) in susceptibility. The authors should discuss this observation and provide explanations for not observing a more relevant resistance phenotype in Fig. 7B (perhaps reduced expression of other receptors does not occur in RNAi experiments contrary to data with cell cultures in Fig. S4?)

-It is interesting that the manuscript presents basically two mechanisms (one trans and one cis) to reduce PxmALP expression. How would two mechanisms resulting in the same phenotype be selected in the same strain? This needs to be discussed.

Reviewer #2: The manuscript entitled ‘MAPK-mediated transcription factor GATAd contributes to Cry1Ac resistance in diamondback moth by reducing PxmALP expression’ deciphered that the expression of PxmALP was cis-regulated by transcription factor GATAd, and lay a solid foundation for comprehensive understanding the mechanisms of Cry1Ac resistance in diamondback moth. The experiments were well designed and scientifically sound, and the manuscript was well written. I recommend the acceptance of the manuscript.

Reviewer #3: Alkaline phosphatase (ALP) is one of the midgut receptors of Bt toxins. This paper identified that PxGATAd directly activates PxmALP expression via interacting with a non-canonical but specific GATA-like cis-response element (CRE) located in the PxmALP promoter region. In addition, PxGATAd is trans-regulated by the upstream activated MAPK signaling pathway. The manuscript was well organized and written. The data are sufficient to support the conclusion in the manuscript.

There are two minor points:

1. lines 339-340, HamALP was mentioned. Are there any reports on the TFs of mALP in other insects? If so, please discuss more about the TFs of mALP in insects.

2. There are a few isoforms of GATA. Here, authors found that GATAd was most important. How about their functions in other related studies.

**Have all data underlying the figures and results presented in the manuscript been provided?**

Reviewer #1: Yes

Reviewer #2: Yes

Reviewer #3: None

PLOS authors have the option to publish the peer review history of their article (what does this mean?). If published, this will include your full peer review and any attached files.

Reviewer #1: No

Reviewer #2: No

Reviewer #3: No

---

## [Decision Letter · Decision Letter 1]

12 Jan 2022

Dear Dr Zhang,

We are pleased to inform you that your manuscript entitled "MAPK-mediated transcription factor GATAd contributes to Cry1Ac resistance in diamondback moth by reducing PxmALP expression" has been editorially accepted for publication in PLOS Genetics. Congratulations!

Yours sincerely,

Subba Reddy Palli, Ph.D.

Associate Editor

PLOS Genetics

Gregory P. Copenhaver

Editor-in-Chief

PLOS Genetics

Comments from the reviewers (if applicable):

Reviewer's Responses to Questions

**Comments to the Authors:**

Reviewer #1: The authors have addressed concerns with previous version

Reviewer #3: no more comments.

**Have all data underlying the figures and results presented in the manuscript been provided?**

Reviewer #1: Yes

Reviewer #3: None

PLOS authors have the option to publish the peer review history of their article (what does this mean?). If published, this will include your full peer review and any attached files.

Reviewer #1: No

Reviewer #3: No

**Data Deposition**

http://datadryad.org/submit?journalID=pgenetics&manu=PGENETICS-D-21-01514R1

**Press Queries**

---

## [Editor Report · Acceptance letter]

31 Jan 2022

PGENETICS-D-21-01514R1 

MAPK-mediated transcription factor GATAd contributes to Cry1Ac resistance in diamondback moth by reducing PxmALP expression 

Dear Dr Zhang, 

We are pleased to inform you that your manuscript entitled "MAPK-mediated transcription factor GATAd contributes to Cry1Ac resistance in diamondback moth by reducing PxmALP expression" has been formally accepted for publication in PLOS Genetics! Your manuscript is now with our production department and you will be notified of the publication date in due course.

With kind regards,

Orsolya Voros

PLOS Genetics

On behalf of:
